# Unlocking the Potential of Kinase Targets in Cancer: Insights from CancerOmicsNet, an AI-Driven Approach to Drug Response Prediction in Cancer

**DOI:** 10.3390/cancers15164050

**Published:** 2023-08-10

**Authors:** Manali Singha, Limeng Pu, Gopal Srivastava, Xialong Ni, Brent A. Stanfield, Ifeanyi K. Uche, Paul J. F. Rider, Konstantin G. Kousoulas, J. Ramanujam, Michal Brylinski

**Affiliations:** 1Department of Biological Sciences, Louisiana State University, Baton Rouge, LA 70803, USA; msing21@lsu.edu (M.S.); gsriva2@lsu.edu (G.S.); xni2@lsu.edu (X.N.); 2Center for Computation and Technology, Louisiana State University, Baton Rouge, LA 70803, USA; lpu1@lsu.edu (L.P.); eejaga@lsu.edu (J.R.); 3Department of Pathobiological Sciences, School of Veterinary Medicine, Louisiana State University, Baton Rouge, LA 70803, USA; bstanf5@lsu.edu (B.A.S.); iuche@lsuhsc.edu (I.K.U.); pjfrider@gmail.com (P.J.F.R.); vtgusk@lsu.edu (K.G.K.); 4Division of Biotechnology and Molecular Medicine, Department of Pathobiological Sciences, School of Veterinary Medicine, Louisiana State University, Baton Rouge, LA 70803, USA; 5School of Medicine, Louisiana State University Health Sciences Center, New Orleans, LA 70112, USA; 6Division of Electrical and Computer Engineering, Louisiana State University, Baton Rouge, LA 70803, USA

**Keywords:** CancerOmicsNet, protein kinases, kinase inhibitors, druggable targets, cancer, explainable artificial intelligence, saliency map

## Abstract

**Simple Summary:**

Protein kinases, which are molecules involved in cell growth and signaling, can go haywire in cancer cells, causing them to multiply uncontrollably. Using drugs to target these kinases shows promise for cancer treatment, but we still have a lot to learn about effectively targeting them. To prioritize kinases to focus on, we developed CancerOmicsNet, an artificial intelligence model that predicts the response of cancer cells to kinase inhibitor treatment. We tested the model extensively and validated its predictions in real experiments with different types of cancer. To understand how the model makes its decisions and the role of each kinase, we used a special tool called a saliency map. This map helps identify the most important kinases driving tumor growth. By examining a wide range of biomedical literature, CancerOmicsNet indeed has shown promise in selecting potential targets for further investigation in various types of cancer.

**Abstract:**

Deregulated protein kinases are crucial in promoting cancer cell proliferation and driving malignant cell signaling. Although these kinases are essential targets for cancer therapy due to their involvement in cell development and proliferation, only a small part of the human kinome has been targeted by drugs. A comprehensive scoring system is needed to evaluate and prioritize clinically relevant kinases. We recently developed CancerOmicsNet, an artificial intelligence model employing graph-based algorithms to predict the cancer cell response to treatment with kinase inhibitors. The performance of this approach has been evaluated in large-scale benchmarking calculations, followed by the experimental validation of selected predictions against several cancer types. To shed light on the decision-making process of CancerOmicsNet and to better understand the role of each kinase in the model, we employed a customized saliency map with adjustable channel weights. The saliency map, functioning as an explainable AI tool, allows for the analysis of input contributions to the output of a trained deep-learning model and facilitates the identification of essential kinases involved in tumor progression. The comprehensive survey of biomedical literature for essential kinases selected by CancerOmicsNet demonstrated that it could help pinpoint potential druggable targets for further investigation in diverse cancer types.

## 1. Introduction

The human genome contains 518 protein kinase genes, 478 of which are classical and 40 atypical [1]. Multicellular organisms live in a complicated environment where signaling routes are vital [2]. Tyrosine kinases mediate cell differentiation, migration, proliferation, programmed cell death, and metabolism. These enzymes catalyze the transfer of a phosphate group from adenosine triphosphate (ATP) to the hydroxyl group of a tyrosine residue on a protein substrate. Cellular homeostasis and communication depend on this covalent post-translational modification [3,4]. Tyrosine kinases play a role in neoplastic progression by regulating cell proliferation and apoptosis, and cancer cells frequently alter these signaling pathways to gain a selective advantage. Aberrant increased signaling from tyrosine kinases gives these enzymes oncoprotein status, causing signaling networks to fail [5]. Human malignancies have been linked to chromosomal reshuffles and genetic mutations that modulate and deactivate lipid and protein kinases and phosphatases [6,7]. Dysregulation of kinases has been shown in neurological, infectious, and immune diseases [8,9,10,11].

In cancer therapy, kinases are the most important drug targets; therefore, the kinome, a set of protein kinases encoded by the human genome, is a key target for cancer treatment. A variety of both synthetic and natural kinase inhibitors are used to treat cancers. The first approved small molecule was fasudil for treating cerebral vasospasms [12]. A quarter of drug discovery research and development projects involve kinase inhibitors. For example, oncogenic kinase drug targets, such as BRAF, EGFR, and PIK3CA, induce tumor cell signaling pathways and contribute to deletions and mutations in PTEN [13,14]. The mutation in the PTEN gene is found in various tumors, including brain, prostate, and breast cancer [15]. Compared to other therapeutic strategies, inhibiting kinase signaling pathways is less cytotoxic to non-cancerous cells, allowing the selective killing of tumor cells with lower toxicity [16,17]. Specific-kinase inhibitors, such as imatinib and dasatinib, produce better results than conventional cytotoxic therapy [18,19]. These kinase inhibitors increased patient survival in gastrointestinal stromal tumors (GIST), and myeloid leukemia (CML). Because of improved clinical efficacy, many small-molecule kinase inhibitors are currently available. These kinase inhibitors improve patient health and clinical outcomes by inhibiting SRC, ABL, mTOR, PDGFRs, VEGFRs, ERBB2, EGFR, and Kit [20,21]. Most of these inhibitors bind to the ATP-binding site [22,23], while others bind to allosteric sites of kinases [24]. Inhibiting kinase activity in treated patients triggers anti-proliferative mechanisms, leading to cancer remission.

Several methods can identify relevant kinases and their cancer-related mutations. The clinical kinase index (CKI) combines a mutational hotspot analysis, overall survival, pathological parameters, and differential gene expression to rank and prioritize clinically significant kinases across 17 solid tumor cancers [25]. The clinical score was calculated by averaging the clinical stage, metastasis (M), lymph node (N), and tumor (T) staging/grade. Kinase expression in each T stage was compared using ANOVA. If kinase is overexpressed between T stages, it has a score of 1. If multiple T stages were significant based on the kinase overexpression, the scores were averaged to a maximum of 1. The M and N stages, histological grade, clinical stage, and pathological stage were evaluated similarly. The final clinical score was a sum of these stages’ scores, with a maximum score that can vary per cancer due to the availability of clinical and pathological data. Then, all parameters were summed up and divided by the maximum possible score and multiplied by 100% to normalize final scores across all cancer cohorts. A kinase only receives a score if its increased expression correlates with stage progression, decreased survival, or significant mutations.

The mutations of kinases in cancer (MoKCa) [26] database has been devised to predict cancer-related protein kinase mutations and annotate them structurally and functionally. To build the database, somatic mutation data from cell lines and tumors were mapped onto protein crystal structures. Mutated amino-acids were highlighted on a sequence-based 3D image of the protein structure and in an interactive package for molecular graphics. Each mutation data point and its functional implications are annotated in the web interface. Proteins are annotated with domains and phosphorylation sites and linked to functional annotation resources. MoKCa provides consistent and coherent assessments and algorithms for each potential cancer-associated mutation to facilitate an authoritative annotation by structural and cancer biologists directly involved in processing and analyzing mutational data.

Despite advancements in the field, there still needs to be a comprehensive study that can identify the structural and evolutionary properties of oncogenic mutations in kinases across the human kinome, regardless of the level of their current understanding. Computational techniques integrating sequence- and structure-based prediction models can be utilized to classify and describe the impact of cancer mutations in protein kinases. It includes an examination of the functional dynamics of protein kinase family members that are recognized to harbor cancer mutations. The predictions resulting from this analysis are then combined with functional dynamics and a collective motion analysis to categorize and elucidate the nature of cancer mutations in protein kinases. Point mutations that activate or inactivate protein kinases can have diverse effects on their activity. Inactivating mutations, for instance, can stabilize the autoinhibitory inactive state, preventing activation or modifying the kinase fold to degrade kinase activity. According to these findings, utilizing the mutational signatures of cancer-causing kinases can assist in designing medications that specifically target mutational hotspots [27]. This approach could pave the way for a new era of precision medicine.

As artificial intelligence (AI) systems are increasingly successful in real-world applications, the complexity of their models is also rising. This has made it incredibly challenging for people to understand the underlying operations and decision-making processes of these intricate models. The decision-making process has now become a “black box” that is impossible to interpret, leading to the need to find a way to comprehend and interpret increasingly complex models. This is crucial in order to progress the model and gain insights into the subject matter at hand. To address this issue, explainable AI (XAI) has been introduced as a collection of methods and tools that aid human users in comprehending and interpreting the results of machine learning algorithms. Various algorithms and methods, such as saliency maps, layer-wise feature map visualization, deconvolution layers [28,29], and complex surrogate and decomposition models, have been suggested for XAI. In this study, we utilize a customized saliency map with varying channel weights to visualize and interpret the decision-making process of the recently developed CancerOmicsNet [30,31,32]. This approach was adopted due to the large size of the cancer data and the complexity of the original model, with the goal to obtain novel insight into drug-cell line interactions identified by CancerOmicsNet.

## 2. Materials and Methods

### 2.1. Deep Learning Model

The base learning model employed in this study has been previously developed to predict the response of a cell line to treatment with a kinase inhibitor in terms of the growth rate 50 (GR50) [30,31]. The input to the model is a graph containing the multi-modal information on gene expression mapped onto the human protein–protein interaction (PPI) network. This model first generates embeddings at the node level of the graph, utilizing the data available at each node and the topological information of the PPI network. The created embeddings are then propagated throughout the graph, using a customized attention mechanism. This process prioritizes more important nodes, such as kinase nodes, containing affinity data, and nodes with significant gene expression, allowing them to propagate more information to their neighbors. The propagation process is iterated multiple times to ensure that the information from the entire graph reaches every node within the graph. Subsequently, a readout mechanism is utilized to summarize the embeddings of all nodes, creating a single embedding that encapsulates the underlying graph information. Finally, a series of fully connected layers are employed to generate the final prediction.

### 2.2. Saliency Graph

To have a better understanding of the underlying role of each kinase in the model, we adopted a common method for image recognition tasks, namely the saliency map. The saliency map is a basic tool in image recognition to explain the input contribution to the output in a trained deep learning model. Since the saliency map can provide information on how the trained model makes decisions, it is an important part of XAI [33,34]. In addition, it can be utilized as a guidance mechanism to improve the efficiency of the training process and achieve more accurate results [35]. In image recognition, the most basic saliency map [33,34] represents the gradient of the output class scores with respect to the input:(1)$$w=\frac{\partial S_{C}}{\partial I}{|}_{I_{0}}$$

The value of the gradient w indicates how much the output score $S_{C}$ changes with the change of the input value at pixel $I_{0}$. For example, if a small change in the input pixel at location (i,j) of channel c, denoted as $X_{i,j,c}$, leads to a significant change in the output class score, this pixel is deemed important in the decision making by the model. Thus, the absolute value of the gradient can be used as a feature importance score. A saliency map M is constructed by calculating the maximum magnitude of the gradient in different channels of the entire image:(2)$$M=\underset{c}{\max}\left|w_{i,j,c}\right|$$

The constructed saliency map is a single-channel image which can be overlayed on top of the original image to visualize the importance scores.

The same concept can be extended to graph-based machine learning. The key difference is that rather than the gradient of the class score against input pixels, the model utilizes the gradient of the class score against input nodes. Although the general idea is similar, the actual implementation of saliency maps is different. Since information propagation is essential for graph neural networks, any change in the node features should be propagated to ensure its effectiveness. Thus, the value of the gradient is calculated as:(3)$$w=\frac{\partial S_{C}}{\partial XA}{|}_{X_{0},A_{0}}$$
where X is the node feature matrix and A is the adjacency matrix ($X_{0}$ and $A_{0}$ refer to the input node).

Another difference between image tasks and graph tasks is that the RGB (red, green, blue) channels for images are of equal importance, whereas channels representing gene expression, node type, and drug binding affinity in cancer-specific graphs are not equally important for the classification. A channel can be thought of as a particular aspect or dimension of the feature vector that carries specific information. While taking maximum values across all channels in image tasks is generally acceptable, it cannot not be employed in graph tasks. To address the different importance of channels in graphs, our model employs a channel weighing scheme [36]. This two-step procedure involves calculating the correlation between the individual channel followed by the weight of each channel. Once the multi-channel saliency scores, denoted as $W=[w_{1},w_{2},\ldots ,w_{c}]$, are computed, the channel correlation matrix is represented by the Gram matrix G, where $G_{k,l}$ is the inner product between the saliency scores. The column-wise average of the resulting Gram matrix is calculated as $V=[v_{1},v_{2},\ldots ,v_{c}]$, where $v_{k}$ corresponds to the average strength of channel k compared to other channels.

Subsequently, a similar procedure to the inverse document frequency [37] is employed to calculate the final channel weight:(4)$$t_{k}=\log\left(\frac{K\epsilon +\sum_{l}v_{l}^{r}}{\epsilon +v_{k}^{r}}\right)$$
where K is the total number of channels, ϵ is a small constant added for the numerical stability, and r is a predefined power-scaling parameter with a value of 2 in our model. The final node saliency map M is then calculated as the matrix product between the initial saliency scores W and the channel weight T:(5)M=W·T

The dimensions of M, W, and T are N×1, N×K, and K×1, respectively, where N is the number of nodes and K is the number of channels (features).

## 3. Results

### 3.1. Architecture and Large-Scale Benchmarks of CancerOmicsNet

Figure 1 shows the architecture of CancerOmicsNet, a graph-based deep learning system employing attention propagation mechanisms to predict the therapeutic effects of kinase inhibitors across various tumors [30,31]. This model integrates multiple heterogeneous data, including biological networks, pharmacogenomics, kinase inhibitor profiling, and gene-disease associations (Figure 1A), to construct a graph specific to the combination of cancer cell lines and drugs (Figure 1B). The graph representation of multi-modal data is processed by a series of graph convolution blocks with a customized attention mechanism (Figure 1C). The attention-based propagation leverages the connectivity of the graph to calculate attention weights for the neighbors of each node. The attention mechanism allows the model to assign different importance levels to the neighbors, based on their relevance to the current node. This enables the model to focus on the most informative nodes during the aggregation process. After obtaining the attention weights, node embeddings are updated by aggregating the information from neighboring nodes, weighted by the corresponding attention scores.

The enhanced embedding (Figure 1D) incorporates the combined impact of the node and its neighboring nodes, resulting in improved representations that capture both local and global contextual information. The graph convolution blocks are applied iteratively, allowing the model to refine the node embeddings through multiple iterations. This iterative process facilitates the progressive integration of information from increasingly distant nodes within the graph, enabling the model to capture higher-order dependencies and global patterns. Following the graph convolution blocks, a graph level embedding is generated using a Set2Set model (Figure 1E). This additional step summarizes all the node embeddings to form a condensed representation of the entire graph, capturing its holistic characteristics and high-level patterns. Finally, the graph embedding is passed through a set of fully connected layers (Figure 1F), which perform further transformations and computations. These layers enable the model to learn complex relationships and dependencies among the graph-level features, ultimately leading to the calculation of the final prediction.

The performance of CancerOmicsNet has been evaluated using a tissue-based cross-validation protocol with the overall area under the curve (AUC) of 0.83 ± 0.02 [30]. In these benchmarks, the entire dataset of 3549 cell line-drug combinations, involving 359 cell lines and 29 drugs, was first divided into groups representing different tissues. Subsequently, a cross-validation was conducted, each time using cancer cell lines from one tissue as a validation set while the remaining cancer cell lines were used for model training. Since those cell lines belonging to different tissues have different gene expression patterns, this procedure ensures that the generalizability of CancerOmicsNet is properly evaluated. Figure 2 shows receiver operating characteristic (ROC) plots for individual tissues. Encouragingly, the performance of CancerOmicsNet is not only high, but it also does not vary significantly for different tissues.

### 3.2. Experimental Validation of CancerOmicsNet

Following the comprehensive benchmarking calculations conducted against a large and diverse dataset [30], several anticancer therapies predicted by CancerOmicsNet have been validated experimentally in live-cell time course assays [32]. Here, we present the results of additional experiments conducted for three cancer cell lines treated with drugs according to the CancerOmicsNet predictions. The same protocol has been used; however, this time, experiments were carried out for multiple initial cell densities that are different from that reported previously. The measured relative cell counts are shown in Figure 3 for cell lines treated with drugs administered at 10 μM, 3.162 μM, and 1 μM concentrations and with the initial cell densities of 312, 156, and 78 cells/well. The treatments of the human triple-negative mammary carcinoma cell line HCC70 with the PI3K/mTOR inhibitor PI-103 (prediction score of 0.74, Figure 3A), and the pancreatic adenocarcinoma epithelial cell line Panc 04.03 with the Src inhibitor PP1 (prediction score of 0.91, Figure 3B) are clearly effective regardless of the initial cell densities. These drugs also exhibit dose-dependent antiproliferative activities as we already reported in the previous paper [32]. The last treatment predicted by CancerOmicsNet of the human prostate cancer cell line DU 145 with the ERK5/BRD4 inhibitor XMD8-92 (prediction score of 0.79, Figure 3C) is effective only at the highest concentration of 10 μM. In addition to the time courses of relative cell counts after drug treatment, the normalized cell counts recorded before and 3 days after treatment are shown in Appendix A. Figure 4 shows fluorescent microscopy images recorded for the treatment of Panc 04.03 cells with PP1. Drug treatments at 10 μM (Figure 4A), 3.162 μM (Figure 4B), and 1 μM (Figure 4C) concentrations are compared to the control group consisting of vehicle (DMSO)-treated cells (Figure 4D). As opposed to the control group that significantly proliferated in 3 days, PP1 effectively inhibited the growth of cancer cells in a concentration-dependent manner. The corresponding microscopy images recorded for the treatment of HCC70 with PI-103 and DU 145 with XMD8-92 are shown in Appendix A, respectively.

### 3.3. Explanation for the Decision-Making Process of CancerOmicsNet

Retrospective benchmarks followed by the experimental validation of CancerOmicsNet demonstrate that the graph-based deep learning system with attention propagation mechanisms can accurately predict the effect of kinase inhibitors on tumor growth across various tissues. In this study, we build a graph-based saliency map to better understand the high accuracy of our model in terms of the underlying roles of individual kinases. These proteins are the most informative components of the cancer-specific molecular networks representing various cell line-drug combinations that are the input to CancerOmicsNet. Figure 5 shows the five top-ranked kinases in various cancer types and their importance scores in other cancers. In addition, the location of these important kinases in the human PPI network is presented in Figure 6 for different cancer types. We analyze the constructed saliency map against the existing biomedical literature, specifically focusing on the significance of top-ranked protein kinases in various cancer types across five tissue categories.

#### 3.3.1. Breast Tissue

The saliency map analysis revealed that the top-ranked kinases for breast cancer are WNK Lysine Deficient Protein Kinase 2 (WNK2), NIMA Related Kinase 11 (NEK11), NIMA Related Kinase 2 (NEK2), Oxidative Stress Responsive Kinase 1 (OXSR1), and Serine/Threonine Kinase 39 (STK39), with the importance scores of 0.247, 0.228, 0.190, 0.174, and 0.171, respectively (Figure 5A and Figure 6A). WNK2 functions as an oncogene, and the decreased activity of this kinase is important in metastasis in breast cancer, glioblastoma, and lung cancer. The WNK2 promoter is regulated by Chromobox Homolog 8 (CBX8), whose overexpression reduces WNK2 expression, resulting in metastasis. A negative relationship between the WNK2 and CBX8 was found in the breast cancer cell line MDA-MB-231 and the insertion of this cancer cell line in a mouse model resulted in increased metastasis [38]. Further, an increased level of MicroRNA 370 (miR-370) plays a role in controlling the activity of the WNK2 gene in breast cancer. When miR-370 was suppressed in breast cancer xenografts, there was an elevation in WNK2 expression, leading to a reduction in cell proliferation and tumor formation [39]. These findings highlight the significance of WNK2 as a crucial target of miR-370 in the context of breast cancer.

A NEK serine/threonine kinase family, comprising 11 members labeled NEK1-NEK11, is essential for cell cycle and microtubule formation. The mRNA level analysis of NEK genes in breast cancer cell lines and breast cancer patients from various data sources revealed that NEK2, NEK4, NEK5, NEK6, NEK8, and NEK11 are overexpressed in breast cancer and may be involved in breast cancer development [40]. The NEK2 gene has been identified as a tumor suppressor gene in breast, thyroid, gastric, and hepatic cancer. Thus, NEK2 is markedly upregulated in human breast cancer tissues, correlating with a poor prognosis, and holds promise as a potential biomarker for identifying breast cancer [41].

OSR1 has been found as overexpressed in various types of cancer, including breast cancer. OSR1 mRNA and protein levels in human breast cancer cell lines are significantly upregulated and the presence of phosphorylated OSR1 in breast cancer patients was identified as an independent indicator of poor prognosis. The overexpression of OSR1 induced the epithelial-to-mesenchymal transition (EMT) in malignant mammary epithelial cells, enhancing their ability to spread. Indeed, eliminating OSR1 in breast cancer cells prevented these phenotypic changes [42]. Overall, OSR1 overexpression is strongly associated with tumor aggressiveness in breast cancer [43]. Disrupting the activity of STK39 through knockdown or inhibitors leads to the destabilization of Snail Family Transcriptional Repressor 1 (SNAI1), thus impairing EMT. This disruption of STK39-SNAI1 interaction was found to decrease tumor cell migration, invasion, and metastasis in breast cancers, both in vitro and in vivo. Consequently, the STK39-SNAI1 interaction is a potential therapeutic target in metastatic breast cancer [44].

#### 3.3.2. Digestive System

Five top-ranked kinases in digestive system cancer shown in Figure 5B and Figure 6B are WNK2 (importance score of 0.342), NUAK Family Kinase 2 (NUAK2, 0.239), NEK3 (0.194), STK39 (0.184), STK17A (0.137), and Dual Serine/Threonine and Tyrosine Protein Kinase (DSTYK, 0.134). The reduced levels of the WNK2 protein, leading to an elevated risk of hepatocellular carcinoma recurrence, can be attributed to somatic mutations and copy number variations. The inactivation of the WNK2 gene, known for its tumor suppressor function, resulted in the activation of the ERK1/2 signaling pathway in hepatocellular carcinoma. This pathway plays a vital role in regulating cell proliferation, differentiation, and survival. Furthermore, the inactivation of WNK2 also led to the infiltration of tumor-associated macrophages, which promoted tumor growth and metastasis. Notably, WNK2 has been identified as a key driver associated with hepatocellular carcinoma in the Chinese population following curative resection [45].

The Hippo-YAP pathway plays a crucial role in regulating cell proliferation, apoptosis, and cell fate. In the context of liver cancer, NUAK2 was identified as a mediator for YAP-induced hepatomegaly and carcinogenesis. Specifically, NUAK2 enhances actin polymerization and myosin activity, thereby maximizing YAP’s activity. In liver cancer, targeting and inhibiting NUAK2 activity results in the decreased growth of YAP-dependent tumors. This highlights the significance of NUAK2 as a potential therapeutic target for controlling YAP-driven tumor growth in liver cancer [46,47]. NEK3 is an essential component in cell migration, proliferation, and viability. Immunohistochemistry and clinicopathological research revealed that NEK3 is overexpressed in gastric cancer samples when compared to their normal counterparts. The elevated level of NEK3 expression is substantially connected with lymph node metastases and a poor prognosis for patients with gastric cancer. Consequently, NEK3 was suggested as a potential target and prognostic marker for gastric cancer [48,49].

Serine-threonine kinases regulate various cellular processes, including cell growth, division, and survival. The dysregulation of serine-threonine kinases can contribute to the development and progression of digestive system cancers. For instance, the increased expression of STK39 in hepatocellular carcinoma (HCC) contributes to a decreased probability of patient survival. This overexpression of STK39 stimulates the PLK1/ERK signaling pathway, resulting in enhanced cell proliferation and metastasis in HCC [50]. STK17A is another member of this family that plays a role in regulating cell death and apoptosis. It has been identified as a tumor suppressor in various cancers. The deregulation or downregulation of STK17A is associated with tumor development, progression, and resistance to therapy. In the context of colon adenocarcinoma and colorectal cancer (CRC), STK17A downregulation has been observed. This decrease in STK17A levels is responsible for the morphological changes observed in adenocarcinoma and CRC, leading to an elevated rate of invasion by cancer cells [51].

DSTYK kinase plays a role in promoting Transforming Growth Factor-β (TGF-β)-induced EMT and the subsequent chemotherapy resistance in CRC cells. Knocking down DSTYK in CRC cells significantly reduces the extent of TGF-β-induced EMT and chemo resistance. Notably, the expression of DSTYK is not only correlated with the survival rate of CRC patients but also positively correlates with TGF-β expression. DSTYK expression was found to be elevated in metastatic CRC samples, with levels higher than those observed in primary CRC samples. These findings underscore the significance of DSTYK in CRC metastasis [52].

#### 3.3.3. Excretory Tissue

The top-ranked kinases for excretory tissue cancer, WNK Lysine Deficient Protein Kinase 3 (WNK3, importance score of 0.284), STK39 (0.210), ROS proto-oncogene 1 (ROS1, 0.133), STK17A (0.123), and nemo like kinase (NLK, 0.121), are presented in Figure 5C and Figure 6C. In papillary renal cell carcinoma (PRCC), the long non-coding RNA (lncRNA), called H19, exhibited overexpression. The expression levels of six target genes regulated by downstream miRNAs, including WNK3, were found to be significantly associated with the prognosis of PRCC [53]. The function of STK39 was investigated in the context of renal cell carcinoma (RCC). It was observed that STK39 exhibited overexpression in RCC tissues, leading to the hypothesis that STK39 may hinder apoptosis to facilitate tumor growth [54]. The examination of the binding model between ROS1 and mitoxantrone, an FDA-approved drug for hormone-resistant prostate cancer (PC) treatment, was conducted. Mitoxantrone acts as an inhibitor of the ROS1 protein, suppressing its phosphorylation and the downstream pathway. This inhibition promotes apoptosis in PC cells, contributing to the therapeutic effect of mitoxantrone [55].

Treatment with the proteasome inhibitor MG132 was found to induce apoptosis in PC-3 cells, and it was observed that this treatment increased the expression of STK17a in these cells, suggesting a potential role for STK17a in mediating the ability of MG132 to induce cell death [56]. Mitogen-activated protein kinases (MAPKs) are enzymes involved in cellular stress responses, cell differentiation, and cell growth, while they also play a crucial role in regulating androgen receptor signaling in both prostate cancer and normal prostate tissue. In prostate cancer metastases, the expression of NLK was found to be reduced, and increasing NLK expression induced apoptosis in cultured cells. Notably, NLK was observed to actively regulate androgen receptor signaling within prostate cancer cells [57]. 

#### 3.3.4. Haematopoietic and Lymphoid Tissue

According to the saliency map analysis, OSR1 (importance score of 0.197), STK39 (0.193), discoidin domain receptor tyrosine kinase 1 (DDR1, 0.189), EPH Receptor A3 (EPHA3, 0.137), and ABL proto-oncogene 1 (ABL1, 0.124) are the five top-ranked kinases in haematopoietic and lymphoid tissue cancer (Figure 5D and Figure 6D). Silencing OSR1 has been demonstrated to enhance the proliferation of acute myeloid leukemia cells while simultaneously suppressing cell death. Moreover, the overexpression of OSR1 leads to the reduced expression of leucine-rich repeat-containing G-protein-coupled receptor 5 (LGR5) and decreased c-Jun N-terminal kinase (JNK) phosphorylation. The abnormal expression of LGR5 in acute myeloid leukemia cells counteracts the effects of OSR1 overexpression, resulting in reduced cell survival and proliferation. Hence, OSR1 exerts a tumor-suppressing effect by inhibiting the LGR5-mediated activation of JNK signaling [58]. The progression of breast and prostate cancer also has been linked to the reduced expression of STK39, and the deletion of STK39 also has implications for B-cell lymphomas. The loss of STK39 leads to an increased cell survival despite DNA damage in B-cell lymphoma and creates a pathway for enhanced resistance in the presence of genotoxic stress [59].

DDR1 is a receptor tyrosine kinase that plays a role in cell adhesion, migration, proliferation, and extracellular matrix remodeling. It is primarily involved in mediating cellular responses to collagen, particularly collagen type I and IV. A high expression of DDR1 in Hodgkin lymphoma cells provides a protective shield against apoptosis, a process that typically occurs in Reed–Sternberg and normal Hodgkin cells. This elevated DDR1 expression is significantly associated with the ZAP70 gene, which is a well-known prognostic marker in chronic lymphocytic leukemia. It is also associated with the time it takes for patients to receive their initial treatment for previously untreated chronic lymphocytic leukemia [60]. EPHA3 plays a crucial role in the angiogenesis of multiple myeloma (MM), and the specific targeting of EPHA3 using a monoclonal antibody has an impact on primary bone marrow endothelial cells (ECs) of MM patients. MMECs exhibit significantly higher levels of EPHA3 expression compared to other ECs. The knockdown of EPHA3 using siRNA resulted in a loss of function, leading to reduced adhesive properties, migration capacities, and tube formation abilities of MMECs in vitro. However, this downregulation did not affect cell proliferation or survival rates. The various angiogenesis-associated functions of MMECs were significantly diminished when EPHA3 was downregulated, emphasizing the importance of EPHA3 as a key component in MM angiogenesis [61].

A fusion gene breakpoint cluster region (BCR)-ABL1 is commonly associated with chronic myeloid leukemia (CML) and a subset of acute lymphoblastic leukemia (ALL). It was observed that BCR-ABL1-*Abl1^−/−^* cells induced an aggressive form of illness resembling the blast phase of CML, while BCR-ABL1-*Abl1^+/+^* cells resulted in a less malignant illness resembling the chronic phase of CML. Both the loss of ABL1 and the allosteric stimulation of ABL1 kinase activity enhanced the antileukemia effects. On one hand, the loss of ABL1 led to the increased proliferation and expansion of BCR-ABL1 murine leukemia stem cells. Therefore, ABL1 can exhibit dual roles by suppressing tumor growth and serving as a potential therapeutic target in leukemias with oncogenic variants of the kinase [62].

#### 3.3.5. Respiratory Tissue

The five top-ranked kinases in respiratory tissue cancer shown in Figure 5E and Figure 6E are ROS1 (importance score of 0.267), WNK3 (0.231), NLK (0.212), OXSR1 (0.196), and WNK2 (0.195). ROS1 plays a significant role in a subset of non-small cell lung cancer (NSCLC) cases. ROS1 gene rearrangements, resulting in fusion with other partner genes, have been identified in a small proportion of lung adenocarcinomas. These ROS1 gene rearrangements lead to the production of abnormal fusion proteins with constitutive kinase activity, which can promote cell growth and survival. ROS1-targeted therapies, such as crizotinib and other ROS1 inhibitors, have demonstrated significant antitumor activity and have become an approved treatment option for patients with ROS1-positive advanced NSCLC. These therapies have shown favorable responses, including tumor shrinkage and prolonged progression-free survival, in ROS1-positive lung cancer patients [63,64].

In patients with NSCLC, WNK3 and IncRNA H19 were found to be upregulated. WNK3 is a target of miR-130a-3p, and there exists a negative correlation between WNK3 and miR-130a-3p expression. The IncRNA H19 is downregulated by acting as a sponge for miR-130a-3p. High levels of WNK3 expression in NSCLC patients are associated with a poor prognosis. The inhibition of WNK3 has been shown to enhance radiosensitivity in NSCLC cells and promote cell death [65]. NLK knockdown cells exhibit a decrease in tumor formation, while NSCLC patients show an increase in NLK expression, which was also associated with the T-stage of the tumor. In in vitro studies, metformin, an oral antidiabetic medication, demonstrated the ability to reduce NLK expression and inhibit cell proliferation in NSCLC cell cultures, indicating its potential as a therapeutic option for NSCLC [66].

Sulfur mustard (SM) exposure induces oxidative stress, affecting the respiratory system. In a study investigating the gene expression profile of lung tissue exposed to SM, OXSR1 showed the most significant elevation. SM-induced lung injury, characterized by disrupted epithelial cells, surfactant malfunction, and in extreme cases, potential lung cancer, can be attributed to the damaging effects of SM on the respiratory system [67]. As mentioned previously in the context of breast cancer, the overexpression of CBX8 has been found to suppress WNK2. Elevated levels of CBX8 have been associated with the reduced expression of WNK2 and enhanced metastasis both in vitro and in vivo across various cancer cell types, including lung cancer [38].

## 4. Discussion

The initial step in a typical drug discovery process involves target identification and prioritization, which can span over a period of up to 20 years and incur substantial costs reaching billions of dollars. This step holds utmost importance as the failure of a drug in clinical settings is often attributed to either inadequate efficacy or the toxicities associated with selecting an inappropriate target [68,69]. The utilization of large-scale omics data can facilitate the target selection process, empowering researchers to assess a protein therapeutic target or biomarker potential by integrating multiple factors. The initial indications of target suitability often arise from analyzing the differential RNA and protein expression between diseased and healthy tissues. This is attributed to the observation that specific genes tend to be expressed at higher levels in diseased tissues than in healthy tissues [70], offering valuable insights into the molecular alterations associated with the disease and helping to pinpoint potential targets for intervention. Additionally, omics data allow for exploring PPI networks and signaling pathways. Key nodes or hubs that play critical roles in disease processes can be identified by mapping the interactions among proteins. These nodes also represent potential targets for drug intervention.

To improve the state-of-the-art in the prediction of the cancer growth rate response to kinase inhibitors, we devised CancerOmicsNet, an AI model utilizing heterogeneous multi-modal data and graph-based techniques. This model takes a graph as input, incorporating gene expression data and PPI networks. The model generates embeddings through node-based information, which are propagated by a customized attention mechanism at the graph level. In this context, nodes deemed more crucial, such as kinase nodes, affinity nodes, and gene expression nodes, play a significant role in transmitting information to their neighboring nodes. Finally, a readout mechanism is employed to combine the embeddings of individual nodes into a single representation, effectively capturing the overall characteristics of the graph. Protein kinases are vital targets in cancer therapy due to their pivotal role in malignant cell signaling. A comprehensive scoring system is necessary to evaluate and clinically rank the relevant kinases. In this study, we employed a customized saliency map with adjustable channel weights to visualize and explain the decision-making process of CancerOmicsNet, enhancing our understanding of the involvement of each kinase. Saliency maps are considered a primary tool in XAI for investigating the inputs contributing to the final prediction outcome.

Traditional AI models, such as deep learning neural networks, often work as black boxes, meaning they generate predictions or decisions without providing transparent explanations about how they arrived at those results. While these models can achieve high accuracy, their lack of explainability can be a significant drawback, especially in critical domains where trust, accountability, and regulatory compliance are essential. XAI aims to address this issue by developing AI systems that provide clear, interpretable, and human-understandable explanations for their actions and decisions. By enabling humans to comprehend and reason about the AI system’s inner workings, XAI enhances AI’s technologies, transparency, trustworthiness, and usability. By generating class scores for each kinase node, the saliency map implemented in CancerOmicsNet assists in identifying the essential kinases that contribute to cancer development and progression. This approach holds significant promise to facilitate further research on these kinases as potential targets for drug development.

## 5. Conclusions

Large-scale omics data offer an opportunity to improve drug discovery by integrating multiple factors and exploring potential therapeutic targets. To advance the prediction of cancer growth rate responses to kinase inhibitors, we developed and experimentally validated CancerOmicsNet, an AI system that utilizes heterogeneous multi-modal data and graph-based techniques. This model effectively captures graph information to generate embeddings and identify crucial nodes for prediction. Notably, protein kinases play vital roles in cancer therapy, and our model provides a comprehensive scoring system to evaluate and clinically rank relevant kinases. Employing a customized saliency map with adjustable channel weights enhanced our understanding of the contribution of individual kinases to cancer development and progression, promoting further research on potential drug targets. Overall, CancerOmicsNet, utilizing XAI technologies, shows significant potential to streamline drug discovery by efficiently identifying target candidates, potentially leading to more effective and safe anticancer therapies.

## Figures and Tables

**Figure 1 cancers-15-04050-f001:**
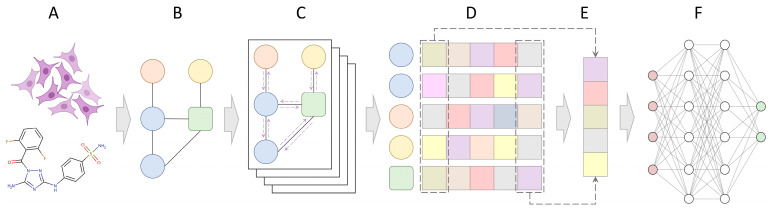
Architecture of CancerOmicsNet. The input for the model consists of multi-modal data for a drug and a cell line (**A**), which are converted to a cancer-specific graph (**B**). A series of graph convolution blocks (**C**) are applied to each node in the graph to generate node embeddings (**D**). Purple arrows represent the attention propagation across the graph. Subsequently, a graph level embedding is generated using a Set2Set model (**E**). Finally, the graph embedding is passed through a set of fully connected layers to make the final prediction (**F**).

**Figure 2 cancers-15-04050-f002:**
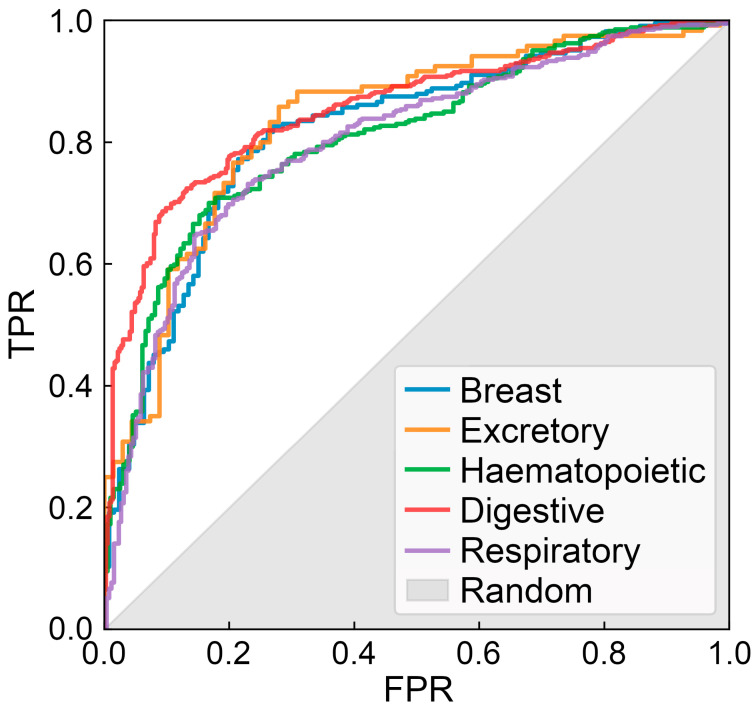
Performance of CancerOmicsNet in predicting the response of cancer cell lines to drugs. The performance is cross-validated at the tissue level for five cancer types: breast tissue, digestive system, excretory system, haematopoietic and lymphoid tissue, and the respiratory system. TPR is the true positive rate, FPR is the false positive rate, and the gray area corresponds to the performance of a random predictor.

**Figure 3 cancers-15-04050-f003:**
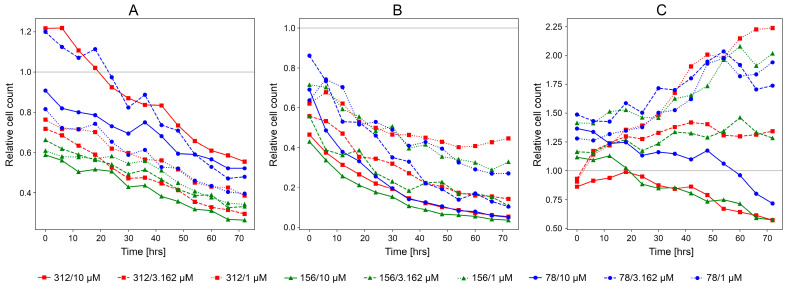
Time courses of relative cell counts after drug treatment at three different initial cell densities (312, 156, and 78 cells/well) and three drug concentrations (10 μM, 3.162 μM, and 1 μM). (**A**) HCC70 treated with PI-103, (**B**) Panc 04.03 treated with PP1, and (**C**) DU 145 treated with XMD8-92.

**Figure 4 cancers-15-04050-f004:**
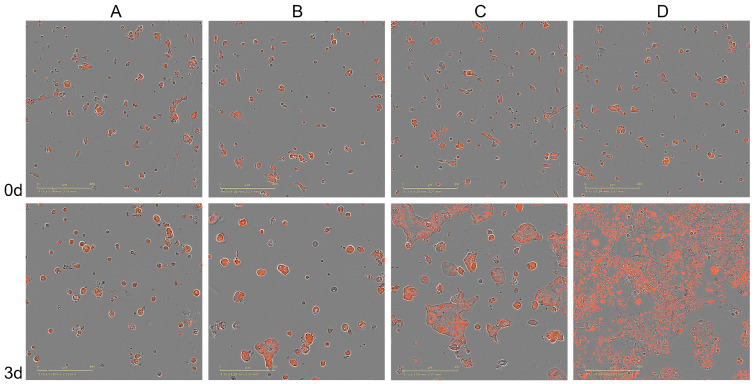
Microscopy images of Panc 04.03 cells treated with PP1. Drug treatment at (**A**) 10 μM, (**B**) 3.162 μM, and (**C**) 1 μM is compared to the vehicle treatment (**D**). The first row shows images recorded before the treatment and the second row shows images taken 3 days after the treatment.

**Figure 5 cancers-15-04050-f005:**
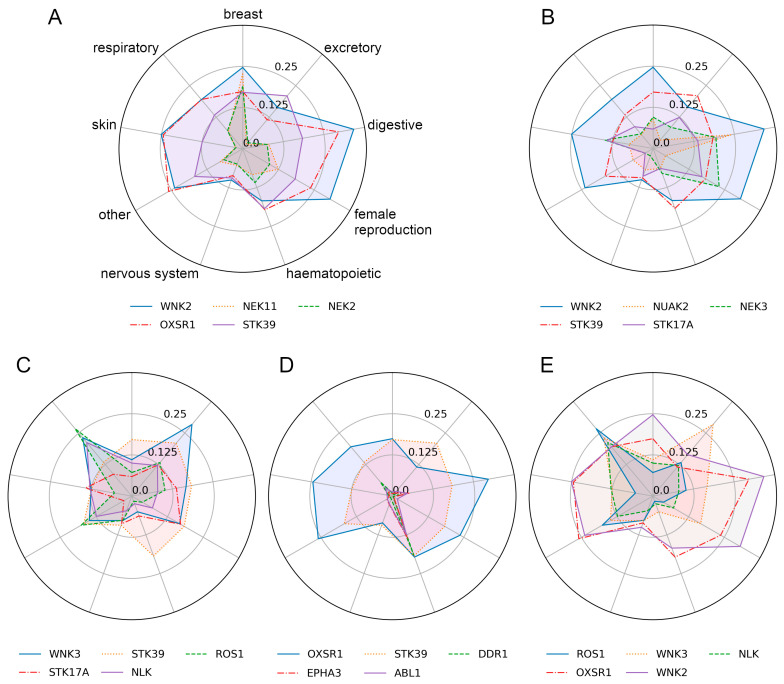
Five top-ranked kinases in five selected cancer types and their importance scores in other cancers. (**A**) Breast tissue, (**B**) digestive system, (**C**) excretory system, (**D**) haematopoietic and lymphoid tissue, and (**E**) the respiratory system. Axes in (**B**–**E**) have the same labels as in (**A**).

**Figure 6 cancers-15-04050-f006:**
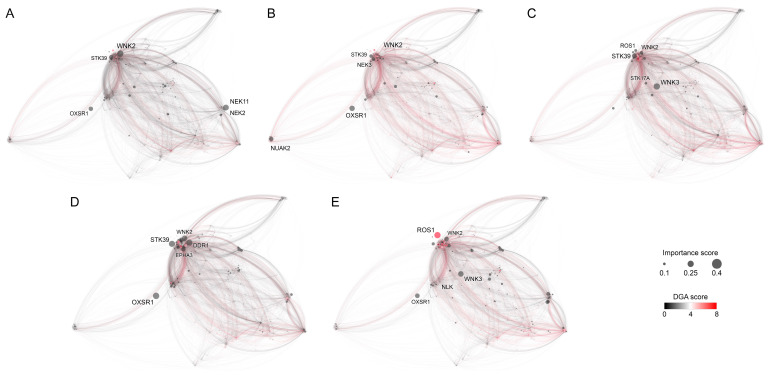
Protein–protein interaction networks for five selected cancer types. (**A**) Breast tissue, (**B**) digestive system, (**C**) excretory system, (**D**) haematopoietic and lymphoid tissue, and (**E**) the respiratory system. The size of kinase nodes corresponds to their importance score, whereas the color represents the disease-gene association score. Five top-ranked kinases for each cancer type are labeled.

## Data Availability

CancerOmicsNet is freely available to the academic community at https://github.com/pulimeng/CancerOmicsNet, accessed on 1 January 2023. All datasets are available at https://osf.io/kv3wa/ and https://doi.org/10.7910/DVN/AMEN3I, accessed on 1 January 2023.

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
