# Peer review of "Unlocking the Potential of Kinase Targets in Cancer: Insights from CancerOmicsNet, an AI-Driven Approach to Drug Response Prediction in Cancer"

_cancers, 2023, doi:10.3390/cancers15164050_

Round 1
Reviewer 1 Report
I would like to congratulate the authors for the portrayed manuscript with the detailed introduction and illustrative results. below are my comments to improve the readability and reproducibility of the work:
1-Adding links to the source code, input data for transparency and ease of replication will be needed a.
2-The experimental validation methods are to be added even into the supplementary.
Other than that, the paper read well and deserves to be published straight away.
Author Response
Our responses are in the attached PDF document.

Reviewer 2 Report
This manuscript reports the Unlocking the Potential of Kinase Targets in Cancer: Insights from Canceromicsnet, an AI-Driven Approach to Drug Response Prediction in Cancer the manuscript is well written it have covered CancerOmicsNet for understanding the role of kinase in the model, saliency map, and AI tool, for the analysis of input contributions to the output of a trained deep-learning model that help in the identification of essential kinases involved in tumor progression.
Author Response

(The authors gave the same response as above.)

Reviewer 3 Report
Deregulated protein kinases are crucial for cancer cell growth and signaling. A comprehensive scoring system called CancerOmicsNet, using artificial intelligence, has been developed to predict the response of cancer cells to kinase inhibitors. The model identifies essential kinases involved in tumor progression and helps identify potential druggable targets for further investigation in different types of cancer. However, there are some confusing points in this article that need to be addressed.
1. The black-box problem has been a significant drawback of artificial intelligence. The authors use saliency map to analyze the decision process of CancerOmicsNet and the contribution of the input to the output, which helps to improve the interpretability of the model. However, the introduction of CancerOmicsNet is not detailed enough, and it would be better to provide a process schematic to introduce how CancerOmicsNet works more clearly.
2. There is limited information regarding CancerOmicsNet. What data sources were utilized for training the model? How were these experiments conducted? Are they comparable to one another? Which parameters were incorporated in the model training process? I suggest that the author present the process of model establishment clearly using charts.
3. To achieve a more comprehensive understanding of the experimental results, it would be advantageous for the author to quantify the experimental data. It is essential to determine the density of cancer cells and the specific method employed by the author for cell counting. Additionally, it is crucial to ascertain how the author determined the growth state after a specified time period, whether it involved cell counting or other alternative methods. Can methods like MTT or CCK-8 be utilized for cell counting? Moreover, it is advisable to represent the experimentally obtained data in a meaningful manner, such as through the utilization of bar graphs, to facilitate a more intuitive and straightforward presentation of the results.
Author Response

(The authors gave the same response as above.)

Round 2
Reviewer 3 Report
The authors have addressed all of my questions, therefore I recommend acceptance of the manuscript.